# NAT10, an RNA Cytidine Acetyltransferase, Regulates Ferroptosis in Cancer Cells

**DOI:** 10.3390/antiox12051116

**Published:** 2023-05-18

**Authors:** Mahmood Hassan Dalhat, Hani Choudhry, Mohammad Imran Khan

**Affiliations:** 1Department of Biochemistry, King Abdulaziz University, Jeddah 21589, Saudi Arabia; 2Centre for Artificial Intelligence in Precision Medicines, King Abdulaziz University, Jeddah 21589, Saudi Arabia

**Keywords:** NAT10, RNA acetylation, ferroptosis, SLC7A11, ROS

## Abstract

Recently, we reported that N-acetyltransferase 10 (NAT10) regulates fatty acid metabolism through ac4C-dependent RNA modification of key genes in cancer cells. During this work, we noticed ferroptosis as one of the most negatively enriched pathways among other pathways in NAT10-depleted cancer cells. In the current work, we explore the possibility of whether NAT10 acts as an epitranscriptomic regulator of the ferroptosis pathway in cancer cells. Global ac4C levels and expression of NAT10 with other ferroptosis-related genes were assessed via dotblot and RT-qPCR, respectively. Flow cytometry and biochemical analysis were used to assess oxidative stress and ferroptosis features. The ac4C-mediated mRNA stability was conducted using RIP-PCR and mRNA stability assay. Metabolites were profiled using LC-MS/MS. Our results showed significant downregulation in expression of essential genes related to ferroptosis, namely *SLC7A11*, *GCLC*, *MAP1LC3A*, and *SLC39A8* in NAT10-depleted cancer cells. Further, we noticed a reduction in cystine uptake and reduced GSH levels, along with elevated ROS, and lipid peroxidation levels in NAT10-depleted cells. Consistently, overproduction of oxPLs, as well as increased mitochondrial depolarization and decreased activities of antioxidant enzymes, support the notion of ferroptosis induction in NAT10-depleted cancer cells. Mechanistically, a reduced ac4C level shortens the half-life of *GCLC* and *SLC7A11* mRNA, resulting in low levels of intracellular cystine and reduced GSH, failing to detoxify ROS, and leading to increased cellular oxPLs, which facilitate ferroptosis induction. Collectively, our findings suggest that NAT10 restrains ferroptosis by stabilizing the *SLC7A11* mRNA transcripts in order to avoid oxidative stress that induces oxidation of phospholipids to initiate ferroptosis.

## 1. Introduction

N-acetyltransferase 10 (NAT10) is a nucleolar protein that promotes cellular events by RNA and protein acetylation [1,2,3,4,5]. Studies have shown that NAT10 regulates DNA damage, senescence, cell proliferation, cell cycle, and apoptosis [5,6,7,8,9,10,11].

Exposing cancer cells to DNA damage agents such as cisplatin and H_2_O_2_ causes an increase in NAT10 protein expression in a dose-dependent manner [5]. Mechanically, during DNA damage, NAT10 is subjected to PARylation through ADP-ribose attachment. The nucleolar PARylation NAT10 gets translocated to the nucleoplasm where it interacts with MORC2 resulting in MORC2 acetylation [7]. In response to the DNA damage, the activated MORC2 initiates DNA damage repair. Collectively, the two studies showed that in response to DNA damage, NAT10 expression increases to initiate DNA damage repair [5,7].

NAT10 mediates the formation of micronuclei (MN) by enhancing DNA replication in cancer cells. The MN is known to regulate the senescence-associated secretory phenotype (SASP) pathway through which NAT10 utilizes the NAT10/cGAS/SASP axis [8].

NAT10 promotes cell proliferation and prevents cell-cycle arrest by acetylation of *BCL-XL* and centrosomal protein 170 (*CEP170*) mRNA transcript [9,10]. NAT10-mediated acetylation of the *BCL-XL* mRNA transcript leads to activation of the PI3K-AKT pathway thereby promoting cell proliferation [9]. NAT10-mediated stability of *CEP170* promotes cell proliferation and chromosomal instability (CIN) [10]. Furthermore, through stable CEP170, NAT10 maintains proper cell cycle distribution [10]. Supporting studies have shown that NAT10 deficiency causes apoptosis and cell-cycle arrest through a decrease in expressions of cyclin-dependent kinase 2 (*CDK2*), *CDK4*, *cyclin D1*, *cyclin E*, and *BCL2*, and an increase in the expression of *p16* and *p21* [11].

NAT10-mediated cell death via apoptosis, either by NAT10 knockdown (KD) or Remodelin treatment as seen in previous studies, has not yielded convincing results as related to a decrease in cell viability and cell morphology. Therefore, we anticipate other cell death types such as pyroptosis, necroptosis, and ferroptosis.

Ferroptosis is a condition characterized by shrunken mitochondria and lipid peroxidation leading to cell death [12]. The basic features of ferroptosis include the availability of redox-active iron, oxidation of polyunsaturated fatty acid (PUFA), and the antioxidant activity of cell membranes against formed lipid hyperoxides [13]. The ferroptosis-induced antioxidant activity depends on three (3) systems, which are as follows: dihydro-orotate dehydrogenase (DHODH)/ubiquinol; ferroptosis suppressor protein 1 (FSP1)/NAD(P)H/ubiquinol; and SLC7A11/glutathione (GSH)/phospholipid hyperoxide (PLOOH) [14]. 

System Xc^-^, containing the SLC7A11/GSH/PLOOH axis, acts as a cystine/glutamate antiport involved in the synthesis of GSH which is converted to oxidized glutathione (GSSG) upon catalytic action of GPX4 [15]. Compounds such as erastin induce ferroptosis by interfering with cystine uptake by inhibiting the System Xc^-^ [16,17,18,19]. Studies have reported the impact of targeting ferroptosis in cancer cells; however, no study has shown whether ferroptosis could be induced via NAT10-mediated acetylation. 

Previously, we discovered that Remodelin, the only known small molecule inhibitor of NAT10, alters lipid metabolic pathways in cancer cells, which include fatty acid elongation in mitochondria, fatty acid metabolism, and mitochondrial beta-oxidation of saturated fatty acids [20]. In line with these findings, we also discovered that NAT10 regulates fatty acid metabolism through ac4C-dependent stability of fatty acid metabolic genes, such as *ELOVL6*, *ACSL1*, *ACSL3*, *ACSL4*, *ACADSB*, and *ACAT1* [21]. Collectively, these studies have shown that NAT10 has a metabolic impact on cancer cells by modulating lipid metabolism. 

The discovery of ferroptosis among the highly modulated pathways from our previous RNA-seq in NAT10-depleted cancer cells led us to postulate that NAT10 depletion derails the expression of mRNAs belonging to key modulators of the ferroptosis pathway in cancer cells. Therefore, in the present study, we identify and assess ferroptosis-related genes that are regulated through NAT10-dependent ac4C RNA modification.

## 2. Materials and Methods

### 2.1. Cell Culture, Transfection, and Treatment

Human breast cancer cells—MCF-7, MDA-MB-231, MDA-MB-468, and T47D—were cultured in Dulbecco’s Modified Eagle Media (DMEM) (BIS BioTech, Jeddah, KSA) with 10% fetal bovine serum (FBS) (Sigma, St. Louis, MO, USA) and 1% penicillin–streptomycin (MOLEQULE-ON^®^, Auckland, New Zealand). All studied cells were maintained in humidified incubator conditions of 37 °C and 5% CO_2_. In this study, cells were grown to 40–60% confluence before transfection. 

NAT10 siRNA transfection was performed using specific siRNA targeting NAT10 (Santa Cruz Biotechnology Inc., Dallas, TX, USA) and Lipofectamine^TM^ RNAiMAX (Invitrogen, Waltham, MA, USA) following the manufacturer’s protocol. Briefly, the stock solution of NAT10 siRNA was prepared to 10 µM and the final concentration of 60 nM of either NAT10 siRNA or scramble control was used for transfection.

Cells were allowed to adhere to plates at 40–60% confluence and treated with Remodelin (Cayman Chemicals, Tallin, Estonia) for 24 h. The half maximal inhibitory concentration (IC_50_) was measured based on the cell viability assay. Both NAT10-depleted and Remodelin-treated cancer cells were administered with 10 µM of Ferrostatin (Fer-1) for 24 h to reverse ferroptosis.

### 2.2. Cell Viability Assay

Cells of 1 × 10^4^ were seeded per well in 96-well plates and subjected to cell viability using MTT (3-[4,5-dimethylthiazol-2-yl]-2,5 diphenyl tetrazolium bromide). Briefly, cells were allowed to adhere to plates for 24 h, followed by NAT10 knockdown or Remodelin treatment. After NAT10 knockdown, a 10 µL solution containing MTT (5 µg/mL, MOLEQULE-ON^®^, Auckland, New Zealand) was added to the wells and incubated for 3 h. After the incubation time was complete, a DMEM medium containing MTT was removed from the wells, 100 µL DMSO was subsequently added to each well and allowed to incubate for at least 30 min at 37 °C. The absorbance of content was read with a microplate reader (BioTek^®^, Winooski, VT, USA) using the Gen5™ software version 1.04 (BioTek^®^, Vermont, USA) for microplate reading and data analysis. Cell viability was graphically represented in mean ± SEM values which were calculated using GraphPad Prism Version 8.0.1(GraphPad, San Diego, CA, USA) [20].

### 2.3. Flow Cytometry

Cells of 1 × 10^6^ were either knocked down with NAT10 siRNA or Remodelin for 24 h, washed with PBS, and trypsinized. 

Lipid peroxidation assay was performed using 2.5 µM C-11 BODIPY (Sigma, Missouri, USA) at 37 °C for 30 min. Cystine-FITC uptake assay was conducted using 2 µM Molecular Biotracker cystine-FITC (Sigma, Missouri, USA) and incubated at 37 °C for 30 min. Membrane integrity-based cell viability assay was performed using 10 µM SYTOX^®^ Green (Invitrogen, MA, USA). Mitochondrial membrane potential was analyzed using 2 µM JC-1 staining (Invitrogen, MA, USA) followed by incubation at 37 °C for 30 min. Reactive oxygen species (ROS) levels were measured by staining cells with 500 nM of CellRox Green (Invitrogen, MA, USA) and then incubated at 37 °C for 30 min.

All assays were acquired at 10,000 cells per sample using Amnis Flowsight. Measurements were performed in technical triplicates and repeated in double biological replicates.

### 2.4. Biochemical Analysis

The cell pellet was washed and resuspended in 200 µL PBS followed by lysis. The cell lysate was then centrifuged at 10,000 rpm for 15 min. The collected supernatant was then subjected to protein quantification using the Bradford method. All samples were normalized before assays.

#### 2.4.1. Glutathione Activity Assay

Glutathione reductase activity was measured using the glutathione reductase assay kit (Millipore-Sigma, GRSA-1KT) following the manufacturer’s instructions. This assay is based on the reduction of GSSG by GR in presence of NADPH. In addition, 5,5′-dithiobis (2-nitrobenzoic acid) (DTNB) reacts with GSH to form 5-thio (2-nitrobenzoic acid). Therefore, the GR activity was measured by the increase in absorbance at 412 nm caused by the reduction of DTNB.

#### 2.4.2. Lipid Peroxidation

Lipid peroxidation was quantified by measuring the malondialdehyde (MDA) formed using the thiobarbituric acid-reactive substances (TBARS) assay according to a previously described method. Briefly, 500 μL of cell lysate was added to 1 mL of TBA:TCA: HCl reagent (0.38% thiobarbituric acid prepared in distilled water and warmed in a water bath, 15% trichloroacetic acid dissolved in distilled water, and 0.25 N hydrochloric acid, ratio 1:1:1), boiled at 90 °C for 20 min, and cooled on ice followed centrifugation at 10,000 rpm for 10 min; the absorbance of TBARS was measured at 532 nm against a reagent blank [22]. 

#### 2.4.3. Superoxide Dismutase Activity Assay

Superoxide dismutase activity (SOD) was determined using an SOD assay kit (19160-1KT-F) following the manufacturer’s protocol. This assay is based on WST-1 [2-(4-Iodophenyl)-3-(4-nitrophenyl)-5-(2,4-disulfophenyl)–2Htetrazolium, monosodium salt] method. WST-1 reacts with superoxide anion to produce a water-soluble formazan. The SOD activity was measured at 450 nm.

#### 2.4.4. Catalase Activity Assay

Catalase activity assay was performed using the ammonium molybdate method. Briefly, 20 µL of the sample was added to 100 µL of 30 mM H_2_O_2_ and allowed to incubate at room temperature for 10 min. The reaction was then stopped with 100 µL 32.4 mM ammonium molybdate, and catalase activity was measured at 405 nm.

### 2.5. Extraction for Metabolomics

Cells were seeded in a 75 cm^3^ flask and allowed to grow for 24 h. At 40–50% confluence, the cells were knocked down with NAT10 siRNA. Cells were then washed with ice-cold PBS and 200 µL of ice-cold distilled water was added to the flask. Cells were collected with a scrapper and vortexed to increase breakage, followed by the addition of methanol and acetonitrile (1:1). Content was incubated at −20 °C for 24 h and centrifuged at 14,000 rpm for 10 min. The supernatant was isolated and taken for metabolomic profiling. 

### 2.6. Untargeted Metabolomics for Lipids Profiling

Metabolomic profiling was performed using LTQ XL™ linear ion trap (Thermo Fisher Scientific, Waltham, MA, USA) LC-MS/MS instrument. 

Briefly, 10 µL of metabolite extracts were injected into the HPLC column (Hypersail gold column C18 Part No: 25005-104630) at 0.250 mL/min flow rate. The metabolite extracts were gradient eluted using mobile phases of A, consisting of 99.9% acetonitrile and 0.1% formic acid (0.1% *v*/*v*), and B, consisting of 0.1% formic acid in 99.9% water (0.1%, *v*/*v*), which constituted the mobile phase using a gradient program in which the component of the solution was varied from 5% to 30% for 30 min, 30% to 50% for 10 min, 50% for 10 min, and finally 50% to 95% for 20 min, with a total running time of 70 min at a column temperature of 30 °C.

For mass spectrometry, several MSn parameters were used: full scanning mode was used, ranging from 80 to 2000 *m*/*z*; helium was used as buffer gas, with nitrogen being used as the sheath gas, with 40 arbitrary units as the flow rate; and a capillary temperature of 270 °C was used with a voltage of 4.0 V, and the spray voltage was set at −3.0 kV.

Metabolite profiles from NAT10-depleted cancer cells were processed using XCMS (https://xcmsonline.scripps.edu/, accessed on 12 June 2022). The processed metabolites were identified with The Human Metabolome Database (https://hmdb.ca/, accessed on 16 July 2022). Metabolomic annotation, which includes principal component analysis (PCA), enrichment analysis, and pathway analysis, were performed with Metaboanalyst 5.0 (https://www.metaboanalyst.ca/, accessed on 10 September 2022).

### 2.7. Total RNA Extraction, cDNA Synthesis and Quantitative RT-PCR

Total RNA was extracted with an RNA isolation kit (Haven scientific, Jeddah, KSA) following the manufacturer’s instructions. The purity and concentration of the extracted RNA were determined at 260/280 nm using a spectrophotometer (Nanophotometer, IMPLEN, München, Germany) and were reverse transcribed into cDNA with a high-capacity cDNA conversion kit (Applied Biosystems, Waltham, MA, USA) using 2 μg of total RNA. Gene expression was determined by qRT-PCR and performed using SYBR Green (Applied Biosystems). The PCR program started with a preheating step of 50 °C for 2 min, 95 °C for 2 min, followed by 40 cycles of denaturation at 95 °C for 15 s, annealing at 60 °C for 1 min, and a final extension at 60 °C for 1 min. Melting curves were obtained at 60 °C. Data were reported as fold change (2^−△△Ct^). Assays were performed independently in triplicate.

### 2.8. ac4C Dot Blot

An RNA dot blot was performed as previously described. Briefly, 6µg of total RNA were placed at 65 °C for 5 min and immediately cooled on ice for 3 min. Denatured RNAs were then spotted on the Hybond N^+^ membrane and crosslinked with UV light. The ac4C modifications on RNA were detected using anti-ac4C (Abcam, Waltham, MA, USA).

### 2.9. RNA Immunoprecipitation (RIP)-PCR

Cell pellets from NAT10 KD cells or Remodelin-treated cells were lysed with an RIP buffer (300 mM Tris-HCl, 150 mM KCl, 0.5 mM fresh DTT, 0.5%(*v*/*v*) NP40, ×1 protease inhibitor) and incubated in −80 °C for 24 h. The lysate was then centrifuged at 14,000 rpm for 10 min at 4 °C, then 10% of the input sample was collected before adding 2 µg of either Anti-ac4C (Abcam, MA, USA) or corresponding anti-IgG (Cell Signaling Technology, Danvers, MA, USA). The content was allowed to incubate overnight at 4 °C with constant gentle rotation. Next, protein A/G agarose beads (Cell Signaling Technology, MA, USA) were added and allowed to rotate for another 2 h. Beads were collected after centrifugation at 1000 rpm for 2 min. The RNA of input and IPed samples were extracted with Tri Reagent^®^ solution (Sigma, Missouri, USA) and subjected to RT-PCR.

### 2.10. mRNA Stability Assay

Cells were seeded in 6-well plates for 24 h followed by treatment with 5µg/mL actinomycin D. Cells were then collected at different time points. Total RNA was extracted with Tri Reagent^®^ solution (Sigma, Missouri, USA) and the gene expressions of the studied genes were determined using RT-PCR. The half-life and turnover rate was calculated according to previous reports [23].

### 2.11. Bioinformatics Analysis

Correlation between NAT10 and ferroptosis genes (SLC7A11 and GCLC) was performed using the cBiorptal database (https://www.cbioportal.org/, accessed on 4 November 2022) in a breast cancer patients’ cohort.

### 2.12. Statistical Analysis

All quantitative data were represented as mean ± SEM. The two-tailed student *t*-test was used for statistical differences between two groups, whereas, for more than two groups, the Tukey multiple comparison test or Sidak’s multiple comparison test were used. All results were considered statistically significant at *p* < 0.05.

## 3. Results

### 3.1. Depletion of NAT10 Induces Ferroptosis in Cancer Cells

Recently, we reported that NAT10 regulates fatty acid metabolism in cancer cells [21]. Among the top enriched pathways we identified from our RNA-seq dataset were fatty acid metabolism and ferroptosis. The RNA-seq dataset was deposited in GEO datasets with accession number GSE210086. Ferroptosis-related genes that were differentially downregulated in NAT10 KD cancer cells included glutamate–cysteine ligase catalytic subunit (*GCLC*), microtubules-associated protein 1 light chain 3α (*MAP1LC3A*), soluble carrier family 7 member 11 (*SLC7A11*) and soluble carrier family 39 member 8 (*SLC39A8*) (Figure 1A,B). Since we found ferroptosis among the top pathways in NAT10 KD cancer cells, we wanted to explore whether NAT10 depletion induces ferroptosis.

We transfected three breast cancer cell lines, namely, MCF7, T47D, and MDA-MB-468 with NAT10 siRNA. We then collected morphological images from knockdown control (siC) and NAT10 knockdown (siNAT10) (Figure 1C, Appendix A). To confirm knockdown, we performed RT-qPCR and ac4C Dot blot, both of which showed a significant reduction in NAT10 mRNA expression and ac4C levels, respectively (Figure 1D,E). Additionally, a decrease in the cell viability (1 day < 2 day < 3 day) of all three cell lines was noticed in NAT10-depleted cancer cells, suggesting NAT10 is crucial for cancer growth (Appendix A).

Expression levels of the ferroptosis-related genes *GCLC*, *MAP1LC3A*, *SLC7A11*, and *SLC39A8* in NAT10 KD cancer cells were significantly downregulated, therefore validating the results from our RNA-seq data (Figure 1F). 

During ferroptosis, depletion of System Xc^-^ (SLC7A11/GSH/PLOOH) generates reduced cystine uptake, glutathione levels, and increased lipid ROS. To explore ferroptosis features in NAT10-depleted breast cancer cells, we performed a cystine uptake assay and observed a remarkable decrease in cystine levels in NAT10-depleted cancer cells, suggesting that cystine uptake could be impaired due to downregulation of SLC7A11 mRNA (Figure 1G, Appendix A). To assess whether the reduction in cystine uptake has an impact on lipid ROS we conducted a C-11 BODIPY assay (Figure 1H, Appendix A). Results from the C-11 BODIPY assay showed a significant increase in lipid ROS levels upon NAT10 depletion in all three cell lines.

Both glutathione reductase (GSR) activity and malondialdehyde (MDA) levels were remarkably altered in NAT10 KD cells. Altogether, the decrease in GSR and increase in MDA levels in NAT10 KD cells provides further evidence of ferroptosis induction through decreased glutathione synthesis and increased lipid peroxidation (Figure 1I–K). 

Taken together, the results provided evidence that NAT10 depletion induces ferroptosis. Since the evidence points toward ferroptosis induction through NAT10 depletion, we then asked whether there is an elevation in fatty acids associated to ferroptosis in NAT10-depleted cancer cells.

### 3.2. Depletion of NAT10 Promotes Excessive Intracellular Production of PUFAs and oxPL

To confirm whether there is an elevation in polyunsaturated fatty acid (PUFA) of phospholipid origin (PLOOH) and oxidized phospholipids (oxPL) in NAT10 KD cells, we performed untargeted metabolomics. Principal component analysis (PCA) showed that NAT10 alters the global metabolomic landscape by comparing the NAT10 KD cells (siNAT10) vs. transfection control (siC) (Figure 2A). 

Identified metabolites showed strong correlation when comparing peak intensities of siNAT10 vs. siC (Figure 2B). Additionally, through enrichment and pathway analysis, the pathways identified were butyrate metabolism, citric acid cycle (TCA cycle), valine, leucine and isoleucine degradation, fatty acid metabolism, and ketone bodies metabolism, which were identified in siNAT10 vs. siC cancer cells (Figure 2C,D).

PLOOH and oxPLs were detected and observed to be highly expressed in siNAT10 compared to siC (Figure 2E). The PL species include cardiolipin (CL), ceramide (Cer), phosphatidate (PA), phosphatidylinositol (PI), and phosphatidylserine (PS). 

CLs are phospholipids predominantly found in the inner membrane of the mitochondrion. CL interacts with proteins from the complexes of electron transport chain (ETC) and oxidative phosphorylation (OXPHOS) to promote proper enzymatic activity and structural integrity [24,25,26]. Studies have shown that high cellular CLs cause lower cellular respiration, decrease ATP and energy wastage, and lower OXPHOS efficiency [24]. 

Ceramides are major components of cell membranes whose overexpression is reported in cancer involving inflammation and cell death [27,28,29]. In particular, an increase in ceramides results in apoptosis, necroptosis, autophagy, and ferroptosis in response to DNA damage agents, anti-cancer drugs, or silencing of an important cancer gene, i.e., NAT10. Additionally, PLs such as PA, PI, and PS and their oxidized products (oxPI and oxPS) have been reported to be highly expressed in ferroptosis [30].

Interestingly, pathways such as the adherens junction, cell cycle, hippo signaling pathway, mTOR signaling pathway, cell senescence, synthesis and degradation of ketone bodies, phosphatidyl inositol signaling pathway, and ferroptosis were identified through joint-pathway analysis using the previous transcriptomics dataset GSE210086 and a metabolomics dataset in metaboanalyst (Figure 2F, Appendix A). The identification of ferroptosis and phosphatidyl inositol signaling pathways from the joint pathway analysis further provided strong evidence that NAT10 deletion induces ferroptosis as well as affects the cell membrane integrity.

Taken together, our metabolomic data suggests an overproduction of PL and oxPLs in NAT10 KD cells correlated with ferroptosis. 

### 3.3. Depletion of NAT10 Induces Oxidative Stress in Cancer Cells

Mitochondria are critical organelles in that they control cellular respiration through electron transport chain coupled oxidative phosphorylation. Abnormal cell proliferation and the survival of cancer cells require a vigorous metabolism, which leads to the accumulation of ROS. To control elevated ROS levels, cancer cells require functional mitochondria and a highly active antioxidant pathway. Therefore, we assessed the impact of NAT10 depletion on mitochondrial function via a mitochondrial membrane potential assay. An increase in JC-1 levels was noticed in the NAT10-deflected cancer cells, which suggests that NAT10 is critical for mitochondrial polarization and proper electron transport (Figure 3A, Appendix A). Mitochondrial membrane damage could cause leakage resulting in the escape of electrons from the intermembrane space to the cytoplasm where they cause the generation of ROS.

Cancer cells evade oxidative stress to promote their survival during progression and metastasis. Additionally, the accumulation of reactive oxygen species (ROS) in cancer cells is lethal and hinders cancer growth and survival.

To confirm if NAT10 depletion affects ROS levels, we measured the ROS levels. High ROS levels were observed in NAT10 KD cancers, suggesting NAT10 is relevant in maintaining ROS levels in cancer cells (Figure 3B, Appendix A). To maintain cellular homeostasis, elevated ROS levels are mitigated by antioxidants such as Superoxide dismutase (SOD) and catalase (CAT) (Figure 3C).

A decrease in SOD and CAT activities were observed in NAT10 KD cells, suggesting a reason why cellular ROS were accumulated in the cytosol (Figure 3D,E). Consistent with previous results, a reduction in the expression of mitochondrial associated respiratory genes such as *NDUFAB1*, *NDUFA3*, and *NDUFA7* was observed in NAT10 KD cells (Figure 3F).

Furthermore, a membrane integrity-based cell viability assay using SYTOX Green showed increased cell death in NAT10-depleted cancer cells, suggesting that the absence of NAT10 in cancer cells results in reduced cell membrane integrity as well as cell viability, which could be due to complication of ferroptosis (Figure 3G, Appendix A).

Taken together, these results demonstrate that ferroptosis is induced in NAT10 KD cells through oxidative stress which provides the required ROS to produce oxPLs due to impaired mitochondrial function and a decrease in SOD and CAT activities.

### 3.4. Depletion of NAT10 Induces Ferroptosis via the SLC7A11/GSH/PLOOH Axis in ac4C Dependent Manner

To further assess the impact of NAT10 as a regulator of ferroptosis, the ac4C RIP-PCR was conducted on all of the four genes, *GCLC*, *MAP1LC3A*, *SLC7A11*, and *SLC39A8*, in NAT10 KD MCF7. Results from the RIP-PCR showed a significant reduction in ac4C levels on the mRNA transcripts of the ferroptosis genes *GCLC*, *SLC7A11*, and *SLC39A8*; however, no effect of ac4C reduction was seen in the *MAP1LC3A* mRNA transcript (Figure 4A, Appendix A). 

To confirm whether a decrease in ac4C levels affects RNA stability, a half-life stability assay was performed. Compared with transfection control (siC), the NAT10 KD (siNAT10) showed a remarkable reduction in the mRNA transcript’s half-life of *GCLC*, *SLC7A11*, and *MAP1LC3A*; however, no significance was recorded for *SLC39A8* (Figure 4B). We did not find connecting evidence of ac4C levels with RNA stability in *MAP1LC3A* and *SLC39A8*; therefore, we presumed that NAT10 regulates ferroptosis through the SLC7A11/GSH/PLOOH axis by maintaining the *SLC7A11* and *GCLC* in an ac4C dependent manner.

To further confirm whether NAT10 depletion induces ferroptosis in cancer cells, ferrostatin-1 (fer-1), a small molecule inhibitor of ferroptosis, was used. Cell viability was measured to determine the half minimum inhibitory concentration (IC_50_) of fer-1 in cancer cells which were 18.45 µM and 19.49 µM for MCF7 and T47D, respectively (Appendix A). Treating NAT10-depleted cancer cells with fer-1 increases cystine uptake in MCF7 and T47D cancer cells (Figure 4C, Appendix A). Additionally, the C11 BODIPY assay showed a remarkable decrease in fer-1-treated NAT10-depleted cancer cells, suggesting reduction in lipid ROS levels (Figure 4D, Appendix A).

Since treatment of NAT10-depleted cancer cells with fer-1 reverses ferroptosis in cancer cells, we then asked whether the same condition is applicable in oxidative stress; therefore, we assessed mitochodrial membrane potential and cell ROS levels in NAT10-depleted cancer cells treated with fer-1. As expected, we noticed a remarkable reduction in JC-1 as well as the cell ROS level, which suggests that the oxidative stress due to NAT10 depletion in the cancer cells is reversed due to inhibition of ferroptosis (Appendix A). 

### 3.5. Remodelin, a Small Molecule Inhibitor of NAT10 Induces Ferroptosis

Previous results showed that NAT10 knockdown induces ferroptosis; therefore, we decided to check whether Remodelin induces ferroptosis. First, the half minimum inhibitory concentration was calculated from results obtained from cell viability of Remodelin-treated MCF7, T47D, and MDA-MB-468. The IC_50_s were 13.42 µM, 36.86 µM, and 21.36 µM for MCF7, T47D, and MDA-MB-468, respectively (Figure 5A). The IC_50_s from the cancer cells were used for their treatment and the cell morphology of cells post-treated with Remodelin was recorded (Figure 5B). Notably, the mRNA expression of *NAT10* in Remodelin-treated cancer cells were significantly downregulated in MCF7 and MDA-MB-468; however, no significant difference was recorded in T47D when comparing Remodelin-treated vs. control (Figure 5C). Expression levels of ferroptosis genes were significantly decreased in Remodelin post-treated cancer cells (Figure 5D). When measuring cystine uptake in Remodelin-treated cancer cells, a significant reduction in intracellular cystine levels was observed, suggesting interference of Remodelin with cystine uptake (Figure 5E, Appendix A). Additionally, lipid ROS was found to be decreased in MDA-MB-468 and MCF7; however, no significant change in lipid ROS levels was recorded in T47D (Figure 5F, Appendix A).

Mechanistically, a reduction in GSR activity and an elevation in MDA levels are the features linked to increased lipid ROS levels; therefore, we assessed MDA levels and GSR activity relative to ferroptosis (Figure 5G). Results showed a relative decrease in GSR activity as well as an increase in MDA levels in Remodelin-treated cancer cells and further confirming ferroptosis induction in Remodelin-treated cancer cells through high levels of lipid ROS (Figure 5H,I). Overall results from Remodelin-treated cancer cells suggest Remodelin interferes with cystine uptake and hence promotes ferroptosis in cancer cells. 

### 3.6. Remodelin Promotes Oxidative Stress in Cancer Cells

A mitochondrial potential assay of Remodelin-treated cancer cells showed an increase in mitochondrial depolarization suggesting Remodelin causes mitochondrial damage and hence could implicate in impaired electron transport chain and oxidative phosphorylation (Figure 6A, Appendix A). The ROS level was assessed in Remodelin-treated cancer cells, and an increase in ROS level was noticed in all three studied cancer cells, thus supporting the evidence of mitochondrial depolarization (Figure 6B, Appendix A). Based on the accumulation of ROS, the activities of SOD and CAT were measured. Results from the SOD and CAT activity showed a significant decrease, suggesting that an increase in ROS levels could be due to decreased activities of SOD and CAT (Figure 6C–E). As in NAT10 KD cells, the Remodelin-treated cancer cells showed a remarkable decrease in the mitochondrial respiratory genes *NDUFAB1*, *NDUFA3*, and *NDUFA7* (Figure 6F). However, no difference was seen in *NDUFA3* expression in MDA-MB-468. This might be attributed to the difference in cell lines.

The SOD and catalase are crucial antioxidant enzymes that scavenge and catalyze ROS when accumulated. Therefore, the depolarized mitochondrial condition that occurs when these enzyme activities are reduced ultimately indicates a high ROS level, which is a driver of ferroptosis. Additionally, we performed a membrane integrity-based cell viability assay using SYTOX Green and, similar to NAT10-depleted cancer cells, the Remodelin-treated cancer cells revealed an increase in cell death compared with the control, suggesting that Remodelin has a crucial impact on cell membrane integrity and reduces the cell viability of cancer cells (Figure 6G, Appendix A).

### 3.7. Remodelin Induces Ferroptosis via Reducing SLC7A11 RNA Acetylation and Stability

Since Remodelin induces ferroptosis and oxidative stress in cancer cells, we next assessed the impact of Remodelin on the ac4C levels and mRNA stability of ferroptosis genes. A significant decrease (*p* < 0.05) in ac4C levels was observed in SLC7A11; however, no significant difference (*p* > 0.05) was observed in GCLC, MAP1LC3A, and SLC39A8 of Remodelin-treated cancer cells (Figure 7A, Appendix A). From the mRNA stability assay, all ferroptosis genes showed a significant decrease in half-life (Figure 7B). 

Taken together, we deduce that only *SLC7A11* is regulated by Remodelin in a dependent manner. The overall results here suggest that Remodelin promotes cystine starvation by preventing the expression of *SLC7A11* through ac4C-dependent RNA modification.

Previous results showed that treatment of NAT10-depleted cancer cells with fer-1 reverses ferroptosis through assessment of cystine uptake and lipid ROS levels; therefore, we asked whether fer-1 could reverse the ferroptosis induced due to Remodelin treatment in cancer cells. Flow cytometric analysis showed consistent increase in cystine uptake after treatment with fer-1 in both MCF7 and T47D; additionally, lipid ROS levels were observed to be decreased in cells after a 24 h treatment with fer-1 (Figure 7C,D, Appendix A). Additionally, 24 h of fer-1 treatment in cancer cells post-treated with Remodelin showed a remarkable decrease in JC-1 levels (cells with depolarized mitochondria) and cell ROS levels, suggesting that fer-1 reverses oxidative stress induced by Remodelin (Appendix A).

Overall, the data here provided further evidence that Remodelin induces ferroptosis in cancer cells.

### 3.8. Clinicopathological Analysis Revealed Positive Correlation between NAT10 with SLC7A11 and GCLC in Breast Cancer

All results of our experiments showed that NAT10 maintains the mRNA stability of SLC7A11 and GCLC through NAT10-mediated ac4C modification; therefore, we asked whether there a correlation between NAT10 with SLC7A11 and GCLC. We performed correlation analysis using cBioportal (https://www.cbioportal.org/) in 10,550 breast cancer patients. Positive correlation was observed between NAT10 vs. SLC7A11 (r = 0.42; *p* = 1.4 × 10^−8^) and NAT10 vs. GCLC (r = 0.33; *p* = 1.5 × 10^−5^), suggesting that both SLC7A11 and GCLC interact with NAT10 in breast cancer patients and, therefore, NAT10 is a promising candidate to target ferroptosis in breast cancer patients (Figure 8A). 

Finally, our results suggest that ferroptosis is induced in both NAT10-depleted and Remodelin-treated cancer cells at different stages of the SLC7A11/GSH/PLOOH axis (Figure 8B). Reduced ac4C levels on mRNA transcript of *SLC7A11* and *GCLC* negatively affect their expression, which in turn prevents cystine uptake and dampened glutathione synthesis, thereby exposing the intracellular environment to higher levels of oxPLs. The oxPLs are overproduced in the presence of high ROS levels due to reduced SOD and CAT activities as well as impaired mitochondrial function. Collectively, our results provide strong evidence of a ferroptotic event in both NAT10-depleted cancer cells and Remodelin-treated cancer cells.

## 4. Discussion

In this study, we discovered that NAT10 depletion induces ferroptosis in cancer cells through the SLC7A11/GSH/PLOOH axis. NAT10 maintains the stability of the mRNA of important ferroptosis pathway genes, namely *SLC7A11* and *GCLC*, through ac4C-dependent RNA modification. Further, we also found NAT10 was crucial for maintaining low levels of oxidized phospholipids that are known executioners of ferroptosis.

Previously, evidence has shown that NAT10 is a critical player in cancer biology by regulating important cellular events such as DNA damage, cell proliferation, cell survival, and senescence [6,7,8,9,10]. Although studies have reported the impact of NAT10 depletion in cancer cells, such as cell morphology, cell viability, and apoptosis, minimal information is reported on the impact of NAT10 on other types of cell death. Therefore, in the present study, we assessed whether NAT10 depletion could induce other types of cell deaths such as necroptosis, pyroptosis, and ferroptosis. Interestingly, we identified ferroptosis among the top enriched pathways from our previous RNA-seq of NAT10 KD cells [21].

Of note, it was recently reported that NAT10 regulates pyroptosis in sepsis through the acetylation of ULK1 RNA in the STING pathway via the ULK1-STING-NLRP3 axis [31]. 

Glutathione is a critical intracellular antioxidant whose function is to detoxify the cell environment of oxPLs and other ROS [15]. SLC7A11 is an important part of the System Xc- whose function is cystine uptake that results in GSH synthesis [16,32,33,34]. The SLC7A11 and GCLC are core regulators of ferroptosis and have been studied as promising targets in cancer [33]. Results from our study strongly suggested that SLC7A11 and GCLC of the SLC7A11/GSH/PLOOH axis are regulated by NAT10-mediated ac4C modification. 

Previously, we discovered that Acyl-CoA synthetase long-chain 1 (*ACSL1*), *ACSL3*, and *ACSL4* were downregulated in NAT10 KD cells [21]. Studies have reported the impact of ACSL4 as an inducer of ferroptosis through the production of Fatty acyl CoA of PUFA (PUFA-CoA), which in the presence of ROS leads to the formation of oxPLs [35,36,37,38,39]. Although we found downregulation of *ACSL4* in NAT10 KD cells, our metabolomics data of NAT10 KD cells showed excessive production of cellular PUFAs and oxPLs. The question is whether other available fatty acyl CoA enzymes are involved in the formation of PUFA-CoA in NAT10 KD cells. Recently, it was reported that ACSL4 promotes cancer progression and, therefore, ACSL4-dependent ferroptosis does not necessarily represent tumor suppression [40]. Another study challenges the current view of ACSL4 as the universal ferroptosis regulator for the following reason: ferroptosis is mostly induced in GPX4-dependent inhibition using RSL3. However, SLC7A11 inhibition with erastin or cystine starvation induces ferroptosis either from GPX4-dependent or GPX4-independent mechanisms; this is because the cystine uptake is not only important for glutathione synthesis but also for biomolecules such as coenzyme A (CoA) [41,42]. The CoA is important for both NAT10 and ACSLs as it is required for acetyl-CoA synthesis, which is the substrate for their enzyme-based catalyzed reactions. Therefore, in order to harmonize our claim of ferroptosis induction by NAT10 and ACSL4 downregulation, we presumed that decreased gene expression of SLC7A11 via NAT10-mediated acetylation and cystine starvation negatively affects CoA production. For example, this can be seen in enzymes that utilize acetyl-CoA, including ACSL4.

We observed increased production of PUFAs and oxPLs in NAT10 KD cells from our metabolomic data. To confirm oxPLs production from our metabolomics data, remarkable reduction in GSR, SOD, and CAT activities, along with elevation in malondialdehyde levels, were observed; this further supports the evidence of hampered glutathione synthesis and ferroptosis induction.

## 5. Conclusions

Collectively, our results have provided evidence that NAT10 depletion induces ferroptosis in cancer cells. They have also provided evidence that the treatment of cancer cells with Remodelin induces ferroptosis and oxidative stress. Therefore, Remodelin causes cell death in cancer mostly by ferroptosis. This could be the reason why studies, even though they have reported the impact of Remodelin on cell proliferation and cell survival, fail to report any significant impact of Remodelin on apoptosis.

In conclusion, our study has unveiled evidence that targeting NAT10 in cancer induces ferroptosis, and we propose that this mechanism could be explored for possible therapeutic and cancer treatments.

## Figures and Tables

**Figure 1 antioxidants-12-01116-f001:**
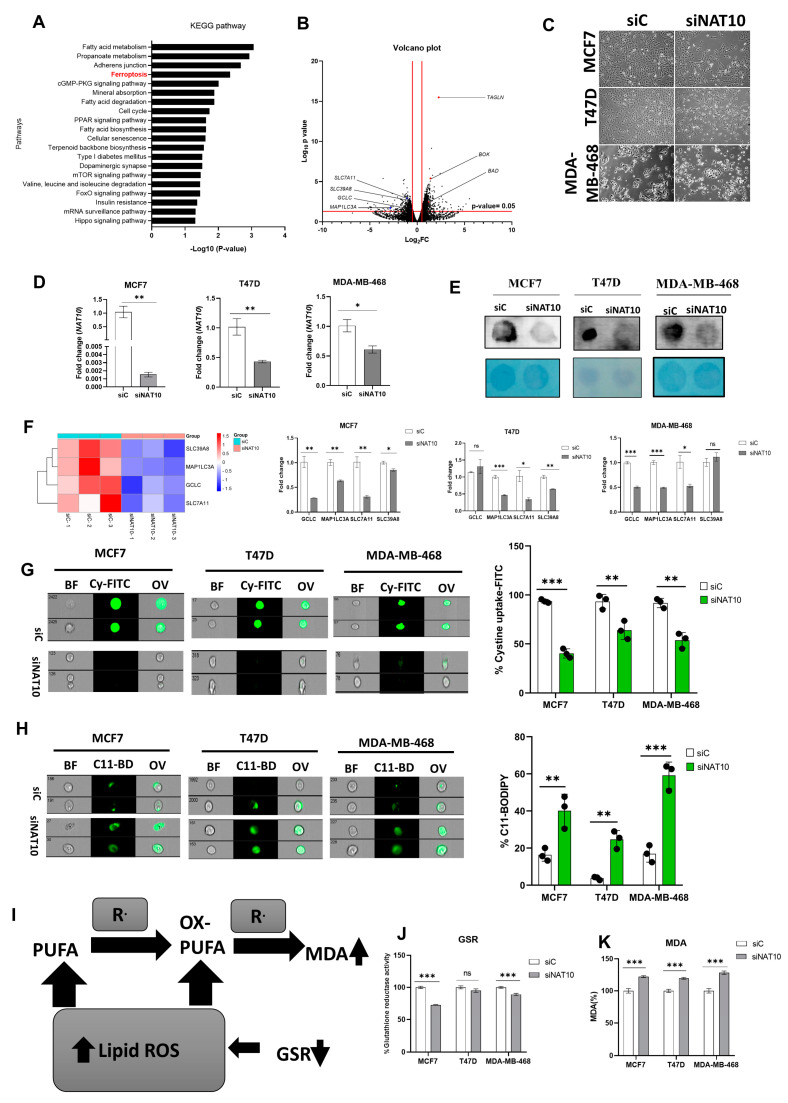
NAT10 knockdown induces ferroptosis in cancer cells. (**A**) Enriched pathways from NAT10 KD cells. (**B**) Volcano plot of differentially expressed genes from NAT10 KD cells, including ferroptosis-related genes. (**C**) Cell morphology of MCF7, T47D and MDA–MB–468 NAT10 KD cells. Images were captured using Nikon microscope at ×10 magnification. (**D**) Expression of NAT10 in NAT10 KD MCF7, T47D, and MDA–MB–468. (**E**) Dot blot image of MCF7, T47D, and MDA–MB–468 NAT10 KD cells, upper panel is the ac4C dot blot and lower panel is the methylene blue stain. (**F**) Expression of ferroptosis-related genes in MCF7, T47D, and MDA–MB–468 NAT10 KD cells. (**G**,**H**) Imaging flow cytometry analysis of cystine uptake and C11-BODIPY assay. (**I**) Mechanistic overview of lipid ROS formation (**J**) GSR activity assay of NAT10 KD cells. (**K**) Lipid peroxidation through malondialdehyde (MDA) measurement in NAT10 KD cells. Data are represented as mean ± SEM (*n* = 3) and *p*-value is represented as * *p* < 0.05; ** *p* < 0.01; *** *p* < 0.001; and ns > 0.05. BF: Brightfield, Cy-FITC: Cystine-FITC; C11-BD: C-11 BODIPY; OV: Overlay.

**Figure 2 antioxidants-12-01116-f002:**
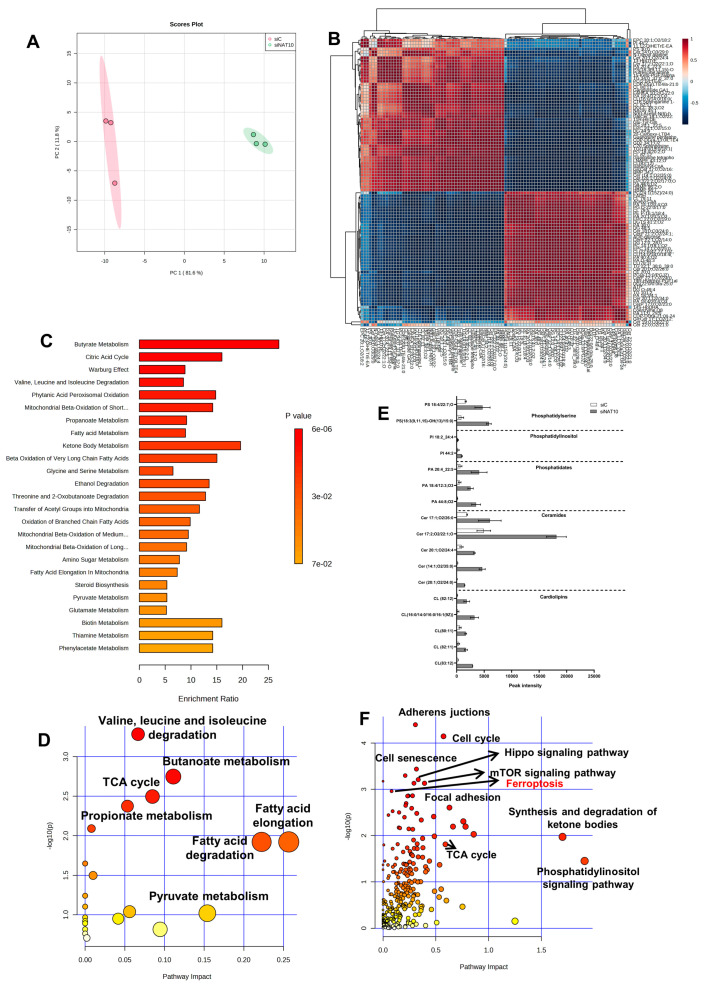
Global metabolic landscape of NAT10 KD cells. (**A**) PCA analysis in siC vs. siNAT10 MCF7 cancer cells. (**B**) Correlation analysis of metabolites identified in siC vs. siNAT10 MCF7 cancer cells. (**C**,**D**) Enrichment and pathway analysis from metabolomes of siC vs. siNAT10. (**E**) PUFAs and oxPLs obtained from metabolomics of siC vs. siNAT10 MCF7 cells. Data are represented as mean ± SEM (*n* = 3) and all results showed significant difference (*p* < 0.05) when comparing transfection control (siC) and NAT10 KD (siNAT10). (**F**) Joint pathway analysis from metabolomes and transcriptomes of NAT10–depleted cancer cells.

**Figure 3 antioxidants-12-01116-f003:**
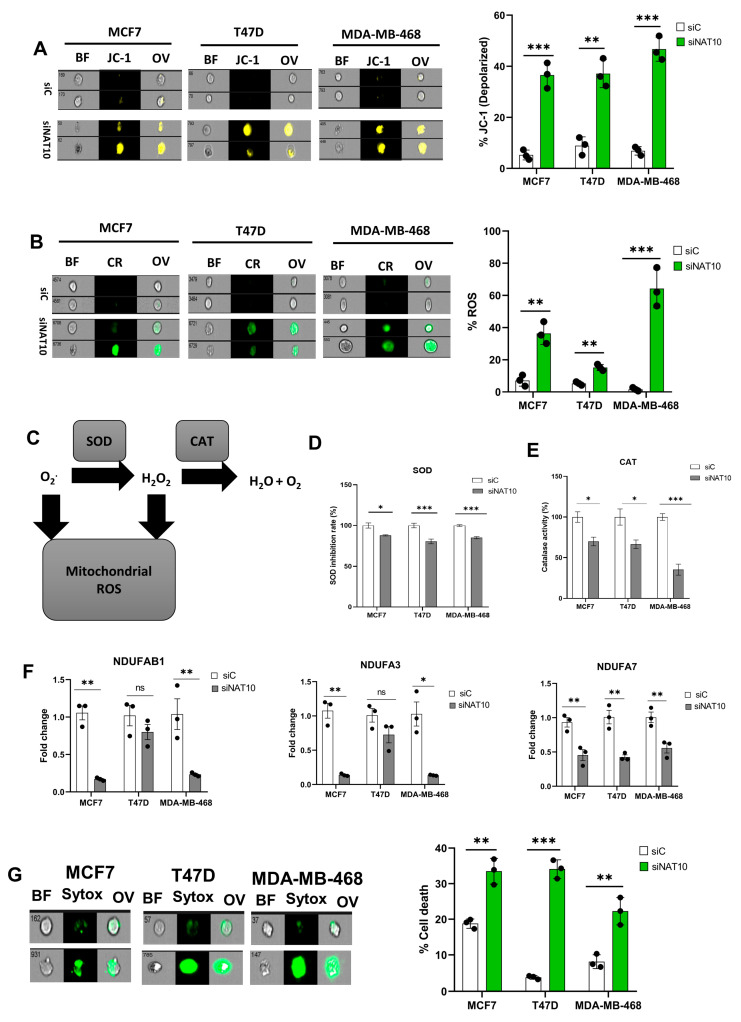
Depletion of NAT10 induces oxidative stress. (**A**,**B**) Imaging flow cytometric analysis of mitochondrial membrane potential and reactive oxygen species (ROS) levels of NAT10 KD cells. (**C**) Mechanistic overview of ROS in cells. (**D**,**E**) SOD and CAT activities of NAT10 KD cells. (**F**) Expression of mitochondrial respiration-related genes. (**G**) Cell membrane integrity and viability determination using SYTOX Green in NAT10-depleted cancer cells. Error bars are represented as mean ± SEM (*n* = 3) and *p*-value is represented as * *p* < 0.05; ** *p* < 0.01; *** *p* < 0.001; and ns > 0.05. BF: Brightfield, CR: CellROX; JC-1; JC-1 stain of cells with depolarized mitochondria; Sytox: SYTOX GREEN; OV: Overlay.

**Figure 4 antioxidants-12-01116-f004:**
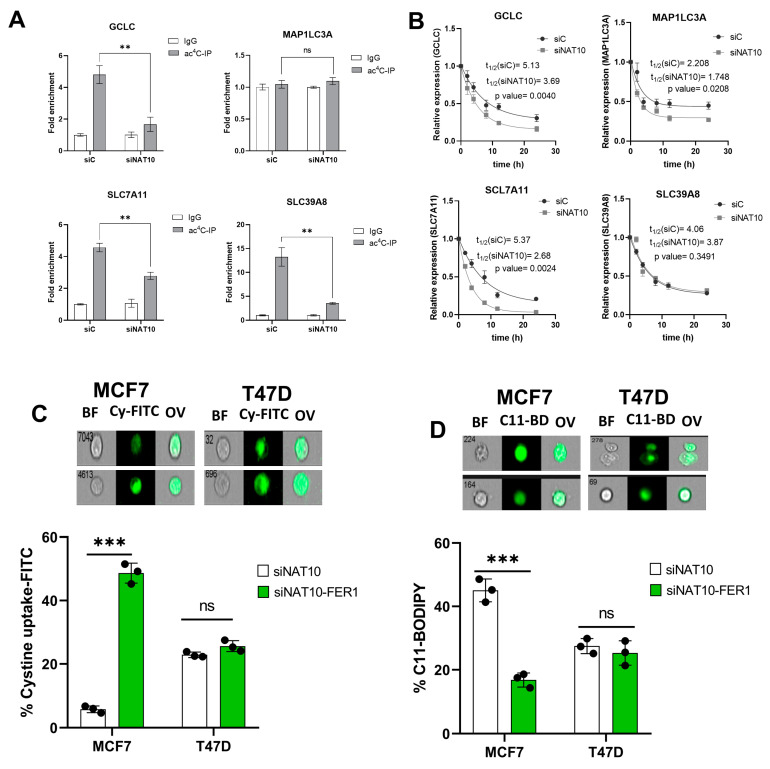
NAT10 maintains the ac4C levels of ferroptosis genes. (**A**) RNA Immunoprecipitation PCR of ferroptosis-related genes. Results are present as mean ± SEM and *p* value ** *p* < 0.01 was considered as statistically significant and ns > 0.05 considered as non-significant. (**B**) RNA stability assay of ferroptosis-related gene analysis was performed using exponential decay method with GraphPad Prism 8.0.1. (**C**,**D**) Flow cytometric analysis of cystine uptake and C11-BODIPY assay in NAT10-depleted cancer cells post-treated with ferrostatin-1 (fer-1) for 24 h. Data are present as mean ± SEM and *p* value *** *p* < 0.001 was considered as statistically significant and ns > 0.05 considered as non-significant. BF: Brightfield, Cy-FITC: Cystine-FITC; C11-BD: C-11 BODIPY; OV: Overlay. Upper panel: qualitative results (cell images). Lower panel: quantitative results.

**Figure 5 antioxidants-12-01116-f005:**
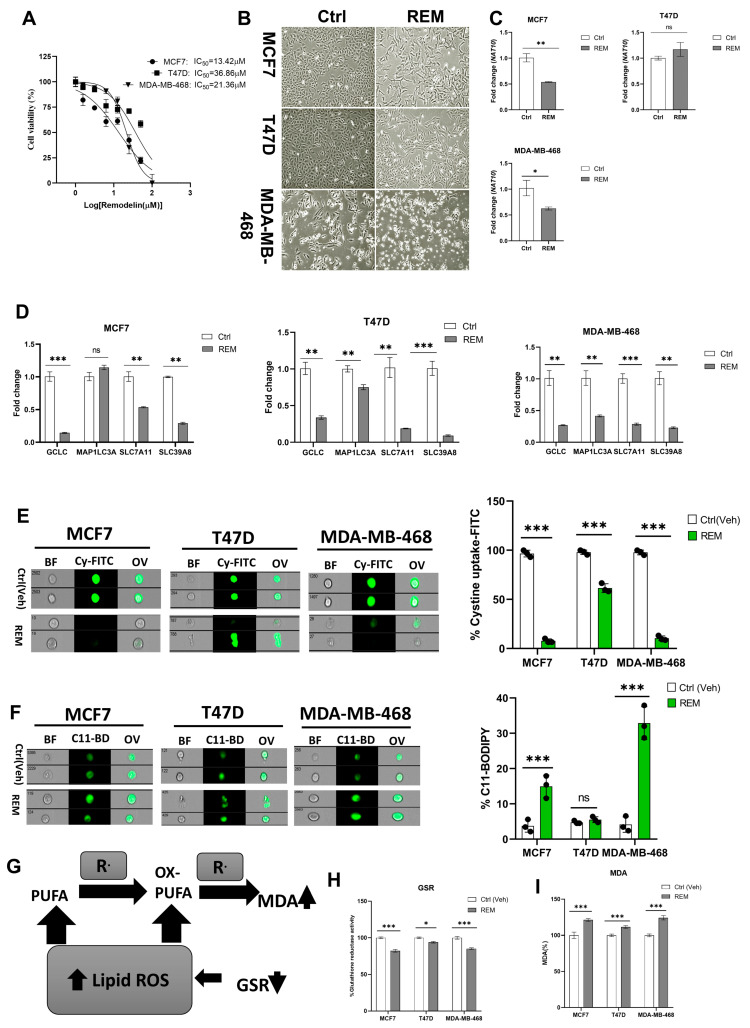
Remodelin induces ferroptosis in cancer cells. (**A**) Cell viability of cancer cells treated with Remodelin showing dose-dependent response at 24 h. Graph was plotted using non-linear regression curve fit using GraphPad prism. (**B**) Cell morphology of MCF7, T47D, and MDA–MB–468 24 h post-treated with Remodelin. Images were captured using Nikon microscope at ×10 magnification. (**C**) Expression of NAT10 in MCF7, T47D, and MDA–MB–468 cancer cells 24 h post-treatment with Remodelin. (**D**) Expression of ferroptosis-related genes in MCF7, T47D, and MDA–MB–468 in Remodelin-treated cancer cells. (**E**,**F**) Imaging flow cytometry analysis of cystine uptake and C11-BODIPY assay of MCF7, T47D, and MDA–MB–468 in Remodelin-treated cancer cells. (**G**) Mechanistic overview of lipid ROS. (**H**) GSR activity assay. (**I**) Lipid peroxidation through MDA measurement of Remodelin-treated cells. Data are represented as mean ± SEM (*n* = 3) and *p*-value is represented as * *p* < 0.05; ** *p* < 0.01; *** *p* < 0.001; and ns > 0.05. BF: Brightfield, Cy-FITC: Cystine-FITC; C11-BD: C-11 BODIPY; OV: Overlay.

**Figure 6 antioxidants-12-01116-f006:**
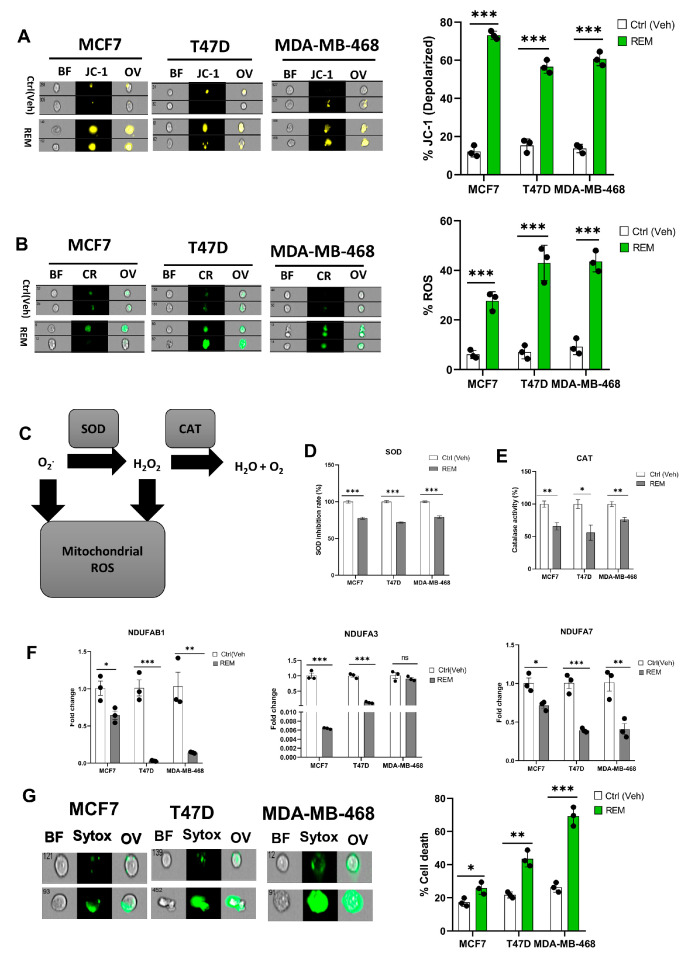
Remodelin induces oxidative stress in cancer cells. (**A**,**B**) Imaging flow cytometric analysis of mitochondrial membrane potential and reactive oxygen species (ROS) levels of Remodelin-treated cancer cells. (**C**) Mechanistic overview of ROS in cells. (**D**,**E**) SOD and CAT activities of cancer cells 24 h after treated with Remodelin. (**F**) Expression of mitochondrial respiration-related genes in Remodelin-treated cancer cells. (**G**) Cell membrane integrity and viability determination using SYTOX Green in Remodelin-treated cancer cells. Error bars are represented as mean ± SEM (*n* = 3) and *p*-value is represented as * *p* < 0.05; ** *p* < 0.01; *** *p* < 0.001; and ns > 0.05. BF: Brightfield, CR: CellROX; JC-1; JC-1 stain of cells with depolarized mitochondria; Sytox: SYTOX GREEN; OV: Overlay.

**Figure 7 antioxidants-12-01116-f007:**
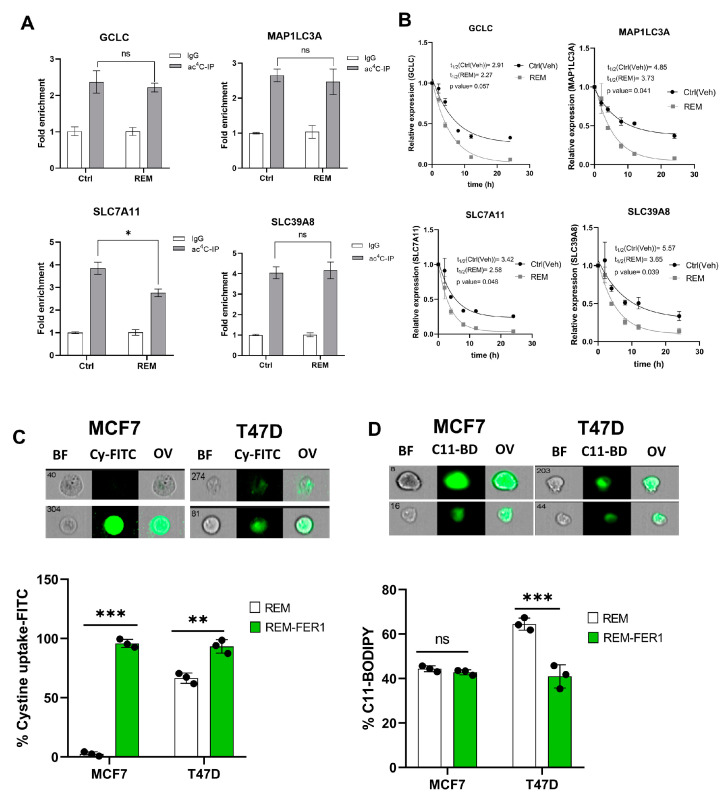
Remodelin induces ferroptosis through ac4C reduction on SLC7A11 mRNA transcript. (**A**) RIP-PCR of ferroptosis-related genes in cancer cells 24 h after treatment with Remodelin. Results are present as mean ± SEM and *p* value * *p* < 0.05 was considered as statistically significant and ns > 0.05 considered as non-significant. (**B**) RNA stability assay of ferroptosis-related gene analysis was performed using exponential decay method with GraphPad Prism 8.0.1. (**C**,**D**) Flow cytometric analysis of cystine uptake and C11-BODIPY assay in Remodelin-treated cancer cells post-treated with ferrostatin-1 (fer-1) for 24 h. Data are present as mean ± SEM and *p* value ** *p* < 0.01; *** *p* < 0.001 were considered as statistically significant and ns > 0.05 considered as non-significant. BF: Brightfield, Cy-FITC: Cystine-FITC; C11-BD: C-11 BODIPY; OV: Overlay.

**Figure 8 antioxidants-12-01116-f008:**
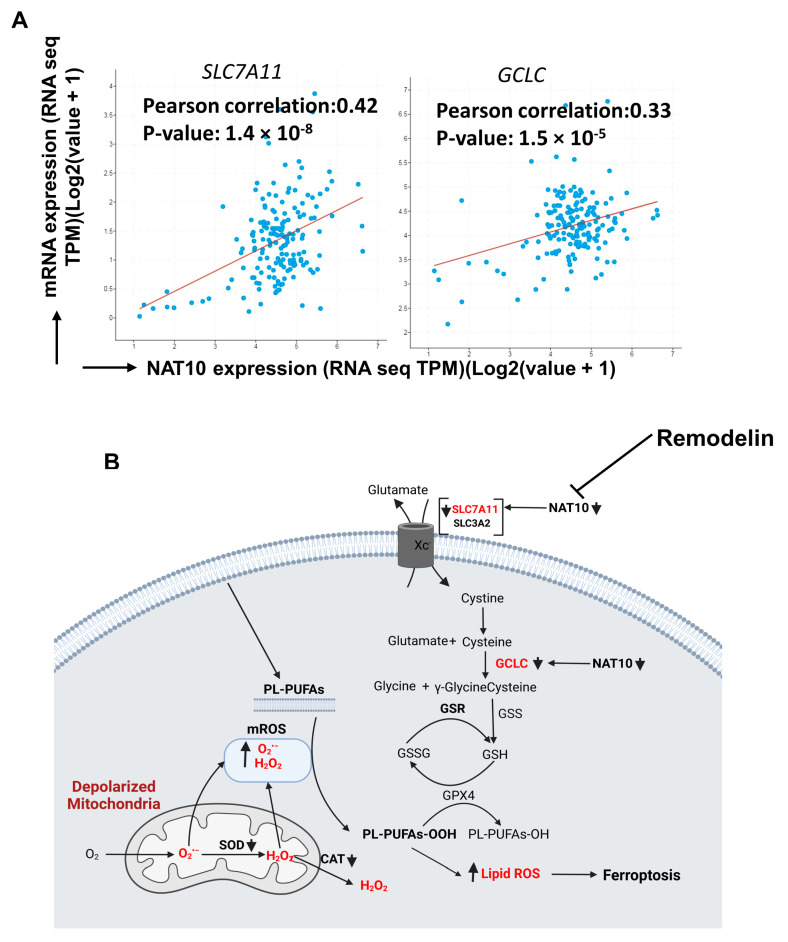
NAT10 regulates ferroptosis in cancer cells. (**A**) Correlation analysis of NAT10 and ferroptotic genes (*SLC7A11* and *GCLC*) retrieved from cBioportal database (https://www.cbioportal.org/, accessed on 26 February 2023). (**B**) Proposed mechanism of NAT10 and Remodelin induction of ferroptosis.

## Data Availability

All data used in this study will be provided upon reasonable request to the corresponding author.

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
