# Peer review of "NAT10, an RNA Cytidine Acetyltransferase, Regulates Ferroptosis in Cancer Cells"

_antioxidants, 2023, doi:10.3390/antiox12051116_

Round 1

Reviewer 1 Report

In this paper Dalhat et al., show that in different cancer cell lines, NAT10 is involved in cell survival and in maintaining low levels of oxidized phospholipids known to mediate ferroptosis. Indeed, they show that mRNA knockdown or pharmacological inhibition of NAT10 leads to Ferroptosis of cancer cells via upregulation of oxidative stress and lipids oxidation.

It is a quite complete and detailed work but some points have to be fixed:

·       Fig. 1E: please explain what are showing the upper and the lower panels

·       Fig 1G: the shape of the cells treated with siNAT10 in bright field looks strange. They look like dead cells, but no the cells that received the same treatment in panel H. Is the uptake of cysteine causing cell death? Is there some problem with siNAT10? Please check and explain.

·       Lines 281-284 related to Fig.1K: MDA level are increased in siNAT10 treated cells in panel K

·       Lines 298-304: please re-write. The last two sentences are not clear it looks like something is missing.

·       Fig.2B: please increase the resolution.

·       Lines 350-351: it is not clear.

·       Line 361: “intracellular space” do you mean cytosol?

·       Please provide a reference for SLC7A11, GCLC, MAP1LC3A and SLC39A8 as ferroptosis related genes.

·       Fig. 4C and D: please indicate what are shown in the upper panel

·       Line 434: please correct “NAT10 knockdown induces Ferroptosis”

·       Line 440: expressions of what? mRNA? Protein? Please specify

·       Line 487-488: please check this sentence: Up to me lipid ROS look increased after Remodelin treatment. Furthermore, inhibition of NAT10 should increase lipid RPS (see Fig. 1).

·       Line 466: please correct “ Figure 6 C-E

·       Line 468: please correct Figure 6F

·       Line 507-511: not clear

·       It would be useful to see a cell viability assay after the different treatment (siNAT10 of Remodelin). And to show that apoptosis is not induced.

Author Response

Reviewer

In this paper Dalhat et al., show that in different cancer cell lines, NAT10 is involved in cell survival and in maintaining low levels of oxidized phospholipids known to mediate ferroptosis. Indeed, they show that mRNA knockdown or pharmacological inhibition of NAT10 leads to Ferroptosis of cancer cells via upregulation of oxidative stress and lipids oxidation.

It is a quite complete and detailed work but some points have to be fixed:

  • Reviewers comment: 1E: please explain what are showing the upper and the lower panels.

Authors response: Panels are explained under the figure 1E caption as suggested by learned reviewer.

  • Reviewers comment: Fig 1G: the shape of the cells treated with siNAT10 in bright field looks strange. They look like dead cells, but no the cells that received the same treatment in panel H. Is the uptake of cysteine causing cell death? Is there some problem with siNAT10? Please check and explain.

Authors response: The cystine uptake experiment in figure 1H is repeated and results are now replaced in figure 1.

  • Reviewers comment: Lines 281-284 related to Fig.1K: MDA level are increased in siNAT10 treated cells in panel K

Authors response: The information was corrected as suggested by learned reviewers.

  • Reviewers comment: Lines 298-304: please re-write. The last two sentences are not clear it looks like something is missing.

Authors response: Sentences are now corrected.

  • Reviewers comment:2B: please increase the resolution.

Authors response: Resolution of figure 2B is improved

  • Reviewers comment: Lines 350-351: it is not clear.

Authors response: The statement is rewritten.

  • Reviewers comment: Line 361: “intracellular space” do you mean cytosol?

Authors response: Yes, we mean cytosol, therefore it is corrected in the text.

  • Reviewers comment: Please provide a reference for SLC7A11, GCLC, MAP1LC3A and SLC39A8 as ferroptosis related genes.

Authors response: SLC7A11 (PMID: 33000412, PMID: 34609966, PMID: 34162423), GCLC (PMID: 33357455, PMID: 36359768), MAP1LC3A (PMID: 35711838, PMID: 36147739), and SLC39A8 (PMID: 34898275).

  • Reviewers comment: 4C and D: please indicate what are shown in the upper panel

Authors response: The upper and lower panels of Figure 4C and D were mentioned as suggested by learned reviewer.

  • Reviewers comment: Line 434: please correct “NAT10 knockdown induces Ferroptosis”

Authors response: The statement is corrected and knockdown is included.

  • Reviewers comment: Line 440: expressions of what? mRNA? Protein? Please specify

Authors response: Expression of mRNA, this has been corrected in text.

  • Reviewers comment: Line 487-488: please check this sentence: Up to me lipid ROS look increased after Remodelin treatment. Furthermore, inhibition of NAT10 should increase lipid ROS (see Fig. 1).

Authors response: The quantitative data is included and information is corrected in Figure 5F.

  • Reviewers comment: Line 466: please correct “ Figure 6 C-E

Authors response: Corrected as Figure 6 C, D to Figure 6 C-E.

  • Reviewers comment: Line 468: please correct Figure 6F

Authors response: Corrected as Figure 6 E to Figure 6 F.

  • Reviewers comment: Line 507-511: not clear

Authors response: The statement is rewritten.

  • Reviewers comment: It would be useful to see a cell viability assay after the different treatment (siNAT10 of Remodelin). And to show that apoptosis is not induced.

Authors response: We have reported cell viability assays of Remodelin (PMID: 34605570) and siNAT10 (PMID: 36149760) previously. Here, we have calculated the Remodelin IC50 through cell viability assay. We are not ruling out apoptosis as a cause of cell death in either Remodelin treated of siNAT10 cancer cells as we have repoted this information previously, however, we are suggesting that the cause of cell death in these treatment is not as a result of apoptosis alone but could be additionally influenced by ferroptosis, therefore we have performed SYTOX green assay to measure the cell death.

Reviewer 2 Report

The manuscript by Khan and co-workers presents an incremental work on their previous results to confirm the role of N-acetyl-transferase 10 (NAT10) as an oxidative-stress regulator in cancer cells. In their previous work, they showed by RNA-seq the upregulation of ferroptosis-related genes upon genetic knock-down (KD) of NAT10. In this work, authors confirm that impairment of NAT10 activity by either  KD or chemicals (remodelin) promotes ferroptosis. They additionally carry out a metabolomic analysis to further charaterise the effects of NAT10 depletion, and compelling oxidative stress analyses. Finally, they make a regression analysis to find a rather poor —despite statistically significant— correlation between NAT10 expression and that of ferroptosis-related genes such as the small ligand carrier 7A11 and glutamate-cysteine ligase. The manuscript is welll organized and written, with minor flaws, and I consider it suitable for publication. Nevertheless I have some concerns and comments to be addressed.

  1. Along the manuscript, authors reject cell death taking place through apoptosis and other PCD pathways, based on RNA-seq and metabolomics. This could be more directly shown using known inhibitors for each kind of PCD. For instance, authors discuss NAT10 being involved in pyroptosis regulation in sepsis. Then I would expect an experiment showing that impairment of Gasdermin D activity does not cell viability.
  2. Related to previous comment, authors carry out experiments using ferrostatin-1—a known ferroptosis inhibitor—but their results are described late, instead of the epigraph 3.1 entitled "Depletion of NAT10 induces ferroptosis [..]"
  3. Although RT-PCR analysis of gene expression is sound, the effects are mediated by the gene products. These are proteins whose amount depend on different factors besides the amount of their encoding RNAs. A quantitation of these products would be welcome, or at least a WB analysis showing the differences between NAT10 KD cells and control ones.
  4. Regarding RIP-PCR experiments, original or representative  RT-PCR curves might be included at least in supplemental material. Same applies to other analyses such as mRNA stability assessment.
  5. Lines 529 to 537, and Figure 8. It is difficult to find a causal relationship from a correlation analysis. Further, Figure 8 indicates correlations between RNA expressions are low,  which may result from indirect effects such as the increase in oxidative stress and signalling from alternative pathways. Moreover, due to the many steps involved in RNA-seq analysis, correlations might be dealt with care. Then, outlawyers could be expectable in the scatter plots. As authors claim they performed their analysis in more than 10,000 samples, I would expect a number of datapoints at least bigger than 2000 in each scatterplot of panel 8A. This may at least add points to justify the lineal regression.
  6. Line 281: According to figure  1K , MDA  levels increase in the KD cells. They are not reduced as the text reads.

Author Response

Reviewer

The manuscript by Khan and co-workers presents an incremental work on their previous results to confirm the role of N-acetyl-transferase 10 (NAT10) as an oxidative-stress regulator in cancer cells. In their previous work, they showed by RNA-seq the upregulation of ferroptosis-related genes upon genetic knock-down (KD) of NAT10. In this work, authors confirm that impairment of NAT10 activity by either  KD or chemicals (remodelin) promotes ferroptosis. They additionally carry out a metabolomic analysis to further charaterise the effects of NAT10 depletion, and compelling oxidative stress analyses. Finally, they make a regression analysis to find a rather poor —despite statistically significant— correlation between NAT10 expression and that of ferroptosis-related genes such as the small ligand carrier 7A11 and glutamate-cysteine ligase. The manuscript is welll organized and written, with minor flaws, and I consider it suitable for publication. Nevertheless I have some concerns and comments to be addressed.

  1. Reviewers comment: Along the manuscript, authors reject cell death taking place through apoptosis and other PCD pathways, based on RNA-seq and metabolomics. This could be more directly shown using known inhibitors for each kind of PCD. For instance, authors discuss NAT10 being involved in pyroptosis regulation in sepsis. Then I would expect an experiment showing that impairment of Gasdermin D activity does not cell viability.

Authors response: We have reported cell viability assays of Remodelin (PMID: 34605570) and siNAT10 (PMID: 36149760) previously. Here, we have calculated the Remodelin IC50 through cell viability assay. We are not ruling out apoptosis as a cause of cell death in either Remodelin treated of siNAT10 cancer cells as we have reported this information previously, however, we are suggesting that the cause of cell death in this treatment is not as a result of apoptosis alone but could be additionally influenced by ferroptosis, therefore we have performed SYTOX green assay to measure the cell death. Additionally to accomodate the learned reviewers suggestion we added cell viability of NAT10 depleted cancer cells shown in supplementary figure S1 B-D.

  1. Reviewers comment: Related to previous comment, authors carry out experiments using ferrostatin-1—a known ferroptosis inhibitor—but their results are described late, instead of the epigraph 3.1 entitled "Depletion of NAT10 induces ferroptosis [..]"

Authors response: We decided to tell our ferroptosis story by showing the induction of ferroptosis followed by inhibition supporting previous results such as Cystine uptake, & C11-BODIPY (Figure 4 C,D and Figure 7 C,D) and Mitochondrial potential & Cell ROX (Supplementary Figure S6 and S11). Ultimately, showing the impact the role of fer-1 in inhibition of ferroptosis and oxidative stress.

  1. Reviewers comment: Although RT-PCR analysis of gene expression is sound, the effects are mediated by the gene products. These are proteins whose amount depend on different factors besides the amount of their encoding RNAs. A quantitation of these products would be welcome, or at least a WB analysis showing the differences between NAT10 KD cells and control ones.

Authors response: We thank the learned reviewer for the input, however, assessment of ac4C levels using Dotblot is a better method to measure NAT10 activity in NAT10 based epitranscriptomic study. Additionally, we have a separate project identifying protein interactors of NAT10.

  1. Reviewers comment: Regarding RIP-PCR experiments, original or representative RT-PCR curves might be included at least in supplemental material. Same applies to other analyses such as mRNA stability assessment.

Authors response: We have included curves in Supplementary Figure S1, S4 and S9.

  1. Reviewers comment: Lines 529 to 537, and Figure 8. It is difficult to find a causal relationship from a correlation analysis. Further, Figure 8 indicates correlations between RNA expressions are low, which may result from indirect effects such as the increase in oxidative stress and signalling from alternative pathways. Moreover, due to the many steps involved in RNA-seq analysis, correlations might be dealt with care. Then, outlawyers could be expectable in the scatter plots. As authors claim they performed their analysis in more than 10,000 samples, I would expect a number of datapoints at least bigger than 2000 in each scatterplot of panel 8A. This may at least add points to justify the lineal regression.

Authors response: We used cBioportal database (https://www.cbioportal.org/) for correlation analysis in human patients and what considered is the level of significance. The learned reviewer might observed that even though the correlation is at 0.42 and 0.33 respectively for SLC7A11 and GCLC respectively, the level of significance make the analysis acceptable.

  1. Reviewers comment: Line 281: According to figure 1K , MDA  levels increase in the KD cells. They are not reduced as the text reads.

Authors response: Statement is now corrected please check Line 286-287.
